# Pre-existing heterosubtypic immunity provides a barrier to airborne transmission of influenza viruses

**Valerie Le Sage**[1], **Jennifer E. Jones**[1], **Karen A. Kormuth**[1¤a], **William J. Fitzsimmons**[2], **Eric Nturibi**[1], **Gabriella H. Padovani**[1], **Claudia P. Arevalo**[3], **Andrea J. French**[1], **Annika J. Avery**[1], **Richard Manivanh**[1], **Elizabeth E. McGrady**[1], **Amar R. Bhagwat**[1¤b], **Adam S. Lauring**[2,4], **Scott E. Hensley**[3], **Seema S. Lakdawala**[1,5]*

1 Department of Microbiology and Molecular Genetics, University of Pittsburgh School of Medicine, Pittsburgh, Pennsylvania, United States of America, 2 Division of Infectious Diseases, Department of Internal Medicine, University of Michigan, Ann Arbor, Michigan, United States of America, 3 Department of Microbiology, Perelman School of Medicine, University of Pennsylvania, Philadelphia, Pennsylvania, United States of America, 4 Department of Microbiology and Immunology, University of Michigan, Ann Arbor, Michigan, United States of America, 5 Center for Vaccine Research, University of Pittsburgh School of Medicine, Pittsburgh, Pennsylvania, United States of America

¤a Current address: Biology Department, Bethany College, Bethany, West Virginia, United States of America
¤b Current address: II-VI Aerospace & Defense in Dayton, Ohio, United States of America
* Lakdawala@pitt.edu

**Data Availability Statement:** All relevant data are within the manuscript and its Supporting Information files.

## Abstract

Human-to-human transmission of influenza viruses is a serious public health threat, yet the precise role of immunity from previous infections on the susceptibility to airborne infection is still unknown. Using the ferret model, we examined the roles of exposure duration and heterosubtypic immunity on influenza transmission. We demonstrate that a 48 hour exposure is sufficient for efficient transmission of H1N1 and H3N2 viruses. To test pre-existing immunity, a gap of 8–12 weeks between primary and secondary infections was imposed to reduce innate responses and ensure robust infection of donor animals with heterosubtypic viruses. We found that pre-existing H3N2 immunity did not significantly block transmission of the 2009 H1N1pandemic (H1N1pdm09) virus to immune animals. Surprisingly, airborne transmission of seasonal H3N2 influenza strains was abrogated in recipient animals with H1N1pdm09 pre-existing immunity. This protection from natural infection with H3N2 virus was independent of neutralizing antibodies. Pre-existing immunity with influenza B virus did not block H3N2 virus transmission, indicating that the protection was likely driven by the adaptive immune response. We demonstrate that pre-existing immunity can impact susceptibility to heterologous influenza virus strains, and implicate a novel correlate of protection that can limit the spread of respiratory pathogens through the air.

## Author summary

Influenza viruses pose a major public health threat through both seasonal epidemics and sporadic pandemics. An individual's first influenza virus infection leaves long-lasting

**Funding:** This work was supported by the National Institute of Allergy and Infectious Diseases (CEIRS HHSN272201400007C, SSL; 1R01AI139063-01A1, SSL; 1R01AI113047, SEH; 1P01AI108686, SEH; CEIRS HHSN272201400005C, SEH). SSL receives additional funding from the American Lung Association (RG-575688) and Charles E. Kaufman Foundation (KA2018-98552) a supporting organization of the Pittsburgh Foundation. Additional funding for JEJ includes T32 AI049820 and the Catalyst Award from the University of Pittsburgh Center for Evolutionary Biology and Medicine. ASL is supported by Burroughs Wellcome Fund PATH award. The funders had no role in study design, data collection and analysis, decision to publish, or preparation of the manuscript.

**Competing interests:** The authors have declared that no competing interests exist.

immunity, which plays an unknown role on susceptibility to airborne transmission of new viral strains. We show that pre-existing heterosubtypic immunity against the 2009 H1N1 pandemic virus protects recipient animals from airborne transmission of a seasonal H3N2 influenza virus, which is independent of cross-neutralizing antibodies. Pre-existing immunity with influenza B viruses was not protective suggesting that this phenomenon is driven by an adaptive response. Taken together, these data indicate that pre-existing immunity is an important barrier to airborne transmission and can influence the emergence and spread of potentially pandemic viruses.

## Introduction

Airborne transmission is essential for the epidemiological success of human influenza A virus (IAV), which imposes a significant seasonal public health burden. Every influenza season is different, with one virus subtype (H3N2 or H1N1) typically dominating and factors such as age, pregnancy and pre-existing medical conditions putting people at increased risk of severe influenza infection. During the 2017–2018 H3N2-predominant season, roughly 79,000 people died in the United States, which is greater than the number that died during the 2009 H1N1 pandemic [1–3]. In the 2017–2018 H3N2 influenza season, 40% of the cases were in the elderly (65+), in contrast to the 2009 H1N1 pandemic in which the highest burden of infection was found in individuals 5–24 years of age (48%) [1,2]. This age-based discrepancy in IAV burden suggests that pre-existing immunity could impact the susceptibility to IAV infection, since people of different age groups are exposed to different strains of IAV in early childhood [4–7].

An individual's first influenza infection typically occurs before the age of five [8] and then most individuals are repeatedly infected with influenza viruses during their lifetime. The antibody response to a person's first influenza infection can be boosted upon exposure with antigenically distinct influenza virus strains in a process classically referred to as 'Original Antigenic Sin' [9]. The impact of pre-existing immunity on the spread of influenza viruses has been understudied, and the few reports in this area have found that, surprisingly, pre-existing immunity against heterologous or homologous strains protects animals against subsequent influenza virus infections [10,11]. In these studies, Steel et al demonstrated that pre-existing seasonal H1N1 or H3N2 immunity in recipient guinea pigs reduced transmission of the 2009 H1N1 pandemic virus [11] and Houser et al observed that pre-existing seasonal H3N2 immunity in donor ferrets prevented transmission of 2011 swine H3N2v virus [10]. Taken together these studies suggests a high barrier to influenza virus infection through the air in the presence of pre-existing immunity. However, these studies used a lag of only 4–6 weeks between primary and secondary infections and exposure times of 7–14 days, which does not mimic human exposure conditions or reinfection time scales. Therefore, additional assessment of pre-existing immunity on susceptibility to infection is warranted.

The majority of published transmission studies use 14 days of continuous exposure in immunologically naïve animals [12–15] and a time interval of 4–6 weeks between primary and secondary infection [10,11]. To address these transmission parameters, we examine the role of timing of exposure and pre-existing immunity to address the barriers to transmission and provide a comprehensive comparison of a seasonal H3N2 virus and the 2009 pandemic H1N1 virus (H1N1pdm09). Ferrets are the preferred models for respiratory droplet transmission because they are naturally susceptible to human isolates of IAV and can transmit infectious IAV particles through the air [13,16,17]. Our results indicate that pre-existing heterosubtypic immunity provides a greater barrier for H3N2 transmission than for H1N1pdm09 virus

transmission, suggesting that pre-existing immunity can drive susceptibility to heterosubtypic infections.

## Results

### IAV transmission to naïve animals is efficient after short and periodic exposures

To investigate the constraint of exposure time on transmission, we examined the most recent IAV pandemic H1N1pdm09 virus (A/CA/07/2009) and a representative human H3N2 virus from the same time period (A/Perth/16/2009). Airborne transmission of these two viruses to naïve ferrets was performed continuously for 7 or 2 days, as well as periodically for 8 hours a day for 5 consecutive days (Fig 1A). H1N1pdm09 transmitted by respiratory droplets to 100% of all naïve recipients at all exposure times. Donor ferrets shed virus in nasal secretions from days 1 to 5 post-infection (Fig 1B, 1C and 1D, red bars), while recipient ferrets had a wider range of shedding (Fig 1B, 1C and 1D, blue bars). Shorter exposure times caused a delay in detectable H1N1pdm09 virus in recipient nasal secretions after day 4 post-exposure (Fig 1C and 1D, blue bars). This observation is consistent with the highly transmissible nature of this virus in multiple transmission systems [15,18]. The H3N2 virus replicated efficiently in donor

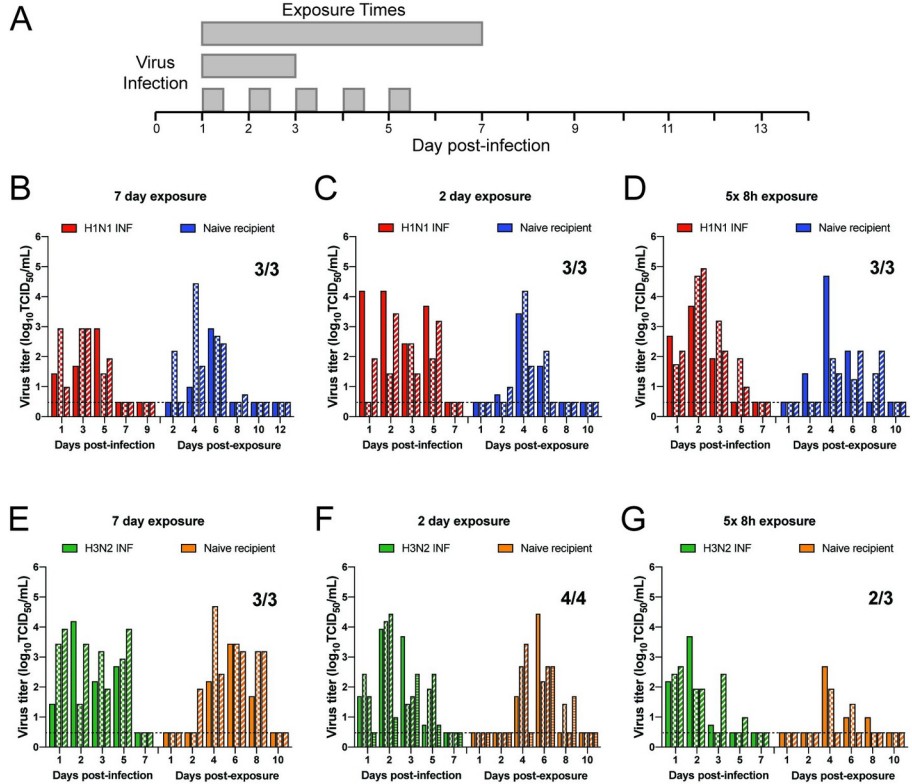

**Fig 1. Efficient transmission of seasonal H1N1 and H3N2 influenza viruses to naïve animals under short or periodic exposures.** (A) Schematic of experimental procedure. Shaded gray bars depict exposure times. Three to four donor ferrets were infected with A/CA/07/2009 (H1N1) (B-D) or A/Perth/16/2009 (H3N2) (E-G) and a naïve recipient ferret was placed in the adjacent cage at 24 hour post-infection for 7 (B and E) or 2 (C and F) continuous days, or 8 hours a day for 5 consecutive days (D and G). Nasal washes were collected from all ferrets on the indicated days and each bar indicates an individual ferret. Limit of detection is indicated by dashed line. Viral titers for donor animals in part G have previously been published in [45].

ferrets on days 1, 2, 3 and 5 (Fig 1E, 1F and 1G, green bars). H3N2 transmitted to 3 out of 3 naïve animals after a 7-day exposure time with slightly delayed shedding kinetics on days 2 to 7 post-infection (Fig 1E, green bars), as compared to ferrets infected with H1N1pdm09 (Fig 1B, red bars). After a 48 hour exposure, H3N2 transmitted to 4 out of 4 naïve recipients (Fig 1F) and 2 out of 3 in a second independent replicate (S1 Fig), with recipients shedding on days 4, 6 and 8 post-infection (Figs 1F and S1, orange bars). At intermittent exposure times transmission of H3N2 virus was slightly reduced to 2 out of 3 naïve recipients, which we still consider to be efficient airborne transmission. Taken together, these results indicate that the A/CA/07/2009 and A/Perth/16/2009 viral strains are highly transmissible to naïve recipients.

## Pre-existing immunity to heterologous strains impacts airborne transmission of influenza viruses at short exposure times

As demonstrated in Fig 1, ferrets are naturally susceptible to human H1N1pdm09 and H3N2 influenza virus infections. To develop a model to mimic sequential influenza infections, we waited between 60 and 84 days between infections to allow for the primary immune response to wane, as shown by others [19–22], and result in two robust infections. Six ferrets were infected with H3N2 virus and then three of those animals were experimentally infected at 60 days post-infection with H1N1pdm09 virus to act as donors (henceforth referred to as 'H3-H1 INF') (Fig 2A). Twenty-four hours after this secondary infection, the three remaining ferrets with H3 pre-existing immunity (S1 Table) were each placed in an adjacent cage to act as recipients (henceforth referred to as 'H3-imm recipient') and exposed for 2 days (Fig 2A). An 84-day gap lead to a robust infection of the H1N1pdm09 virus in experimentally inoculated H3-H1 INF donors with virus detected in ferret nasal secretions on multiple days (Fig 2B). We observed that transmission of H1N1pdm09 to recipients was not greatly impacted by pre-existing H3N2 virus immunity as 2 out of 3 H3-imm recipients became infected after a 2-day exposure period (Fig 2B). Interestingly, the duration and kinetics of H1N1pdm09 shedding in donors and recipients were different between naïve animals and H3-imm animals (Fig 1C vs Fig 2B).

In a complementary study, ferrets with pre-existing H1N1pdm09 immunity were experimentally infected with H3N2 virus to act as donors (henceforth referred to as 'H1-H3 INF') in a subsequent transmission experiment 84 days later (Fig 2A). Robust replication of H3N2 in H1N1pdm09 immune donors (H1-H3 INF) was observed (Fig 2C, green bars) and recipients with pre-existing H1N1pdm09 immunity (henceforth referred to as 'H1-imm recipient') were placed in each adjacent cage 24 hours post-infection. The recipients were exposed to the H1-H3 INF donors for 2 continuous days. Surprisingly, no virus was detected in the nasal secretions of the H1-imm recipients (Fig 2C, orange bars) and no seroconversion was observed on day 13 post-exposure (S1 Table). We previously demonstrated that H3N2 was able to transmit to 6 out of 7 recipients with no prior immunity (Figs 1F and S1). In comparison to this efficient transmission, pre-existing H1N1pdm09 immunity resulted in a complete block to H3N2 transmission during a 2-day exposure window (Fig 2C).

To discern whether donor or recipient immunity was critical for the blockade in H3N2 virus transmission, we examined the spread of H3N2 virus when either the donors or recipients were H1-imm, but not both. In Fig 2D, H1-H3 INF donors transmitted the virus to 2 out of 4 recipients without prior immunity (50%) (Fig 2D, orange bars). This H3N2 transmission efficiency was reduced as compared to 85% (6 out of 7) in animals without prior immunity (Figs 1F and S1), indicating that donor immunity may partially contribute to a barrier in H3N2 transmission. In contrast, 0 out of 4 H1-imm recipients became infected after a 2-day exposure to donors infected with H3N2 virus as their primary infection (Fig 2E, orange bars),

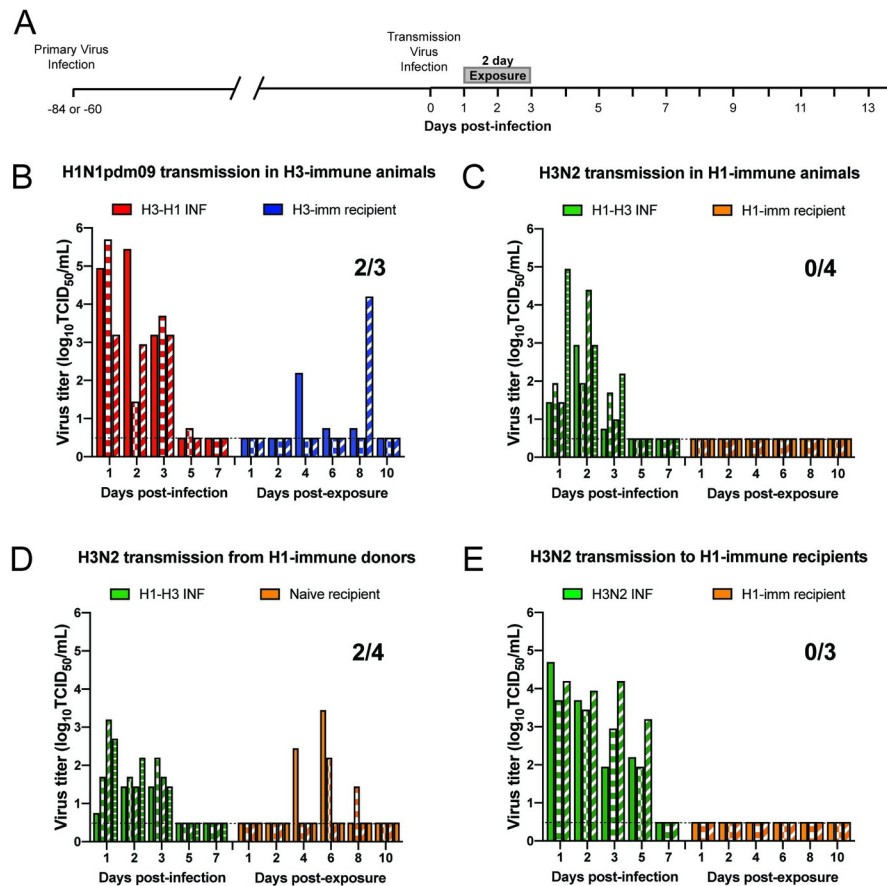

**Fig 2. Pre-existing heterosubtypic immunity blocks H3N2 transmission but not H1N1pdm09 transmission.** (A) Schematic of experimental procedure, gray bars depict 2-day exposure time. (B) All donor and recipient animals were infected with A/Perth/16/2009 (H3N2) virus 60 days prior to H1N1pdm09 (A/CA/07/2009) transmission. Donor animals are denoted as H3-H1 INF, and recipient animals are 'H3-imm'. (C) All donor and recipient animals were infected with A/CA/07/2009 (H1N1pdm09) virus 84 days prior to H3N2 (Perth/16/2009) transmission. Donor animals are denoted as H1-H3 INF, and recipient animals are 'H1-imm'. The contribution of donor and recipient immunity was tested separately for A/Perth/16/2009 (H3N2) transmission. Donors only (D) or recipient only (E) had pre-existing immunity to A/CA/07/09 H1N1pdm09. Donor animals are denoted as H1-H3 INF or H3N2 INF (no prior immunity), and recipient animals are either 'naïve' to indicate no prior immunity or 'H1-imm'. Each bar represents an individual animal. Limit of detection is denoted by a dashed line.

which suggests that heterosubtypic immunity in recipients was sufficient to block H3N2 virus transmission. Taken together, these results indicate that airborne transmission of H3N2 cannot overcome the barrier imposed by H1N1pdm09 immunity but transmission of H1N1pdm09 can overcome H3N2 immunity even within short exposure times.

## Barrier to H3N2 airborne transmission is independent of neutralizing antibodies

To elucidate the difference in protection provided by H1N1pdm09 versus H3N2 immunity, we first compared the shedding of the naïve and immune donors. Higher viral loads in the donors might be expected to result in more efficient transmission, yet they do not always correlate with each other [23]. The H1N1pdm09 and H3N2 nasal washes were titered at different times on different cell lines. To directly compare shedding kinetics between transmission experiments, we quantified the viral RNA levels in nasal washes of H1 or H3 infected donors,

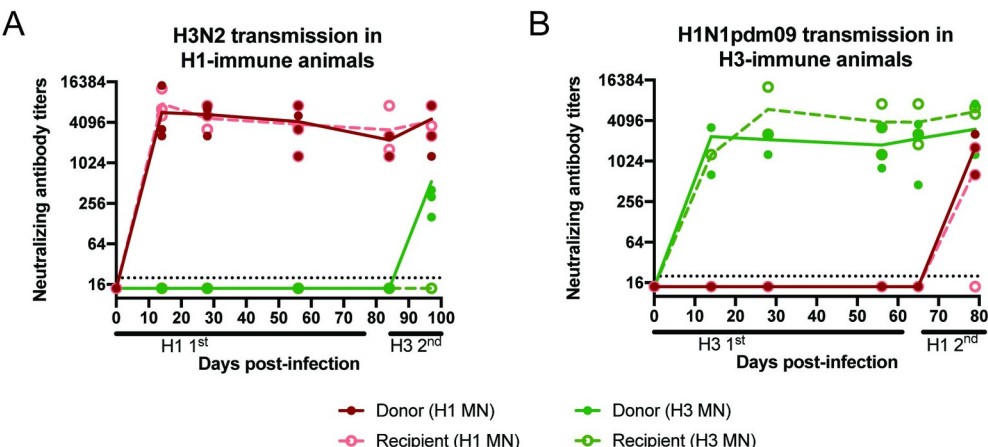

**Fig 3. Neutralizing antibody titers against H1N1 and H3N2 viruses.** We examined the neutralizing antibody titers against A/CA/07/09 (H1N1pdm09; red lines) and A/Perth/16/09 (H3N2; green lines) from ferrets depicted in Fig 2B and 2C. Sera from animals from Fig 2C; H1N1pdm09 primary infection followed by H3N2 transmission are depicted in panel A. Sera from animals from Fig 2B, H3N2 primary infection followed by H1N1pdm09 transmission, are depicted in panel B. At the indicated times, serum was collected and neutralizing antibodies against both H1N1pdm09 (red lines) and H3N2 (green lines) were determined by microneutralization assay. Donor infected animals are denoted in solid circles and recipient animals are denoted by open circles. Each point is an individual animals and trend lines for donor (solid) or recipient (dashed) animals are shown. Limit of detection is indicated by the dashed line.

which revealed similar levels between these two viral infections during both primary and secondary infections (S2A and S2B Fig). These results show that the block in H3N2 transmission was not due to significantly decreased viral replication by ferrets with pre-existing H1N1pdm09 immunity.

To determine whether the H1-imm recipients were generating cross-protective H3 neutralizing antibodies, we performed neutralization assays with sera from animals after primary or secondary infection. Sera from the H3-imm recipients and H3-H1 INF donors exhibited robust H3N2 neutralizing antibodies that persisted from day 14 post infection/exposure and appeared to increase slightly upon heterologous challenge (Fig 3B). Sera from the H3-imm recipients had no detectable cross-reacting neutralizing antibodies against H1N1pdm09 (Fig 3B), with the H1N1pdm09 virus transmitting efficiently to 2 out of 3 H3-imm recipients (Fig 2B). Similarly, sera from H1-imm ferrets produced strong H1N1pdm09 neutralizing antibody titers that waned slightly over 84 days and increased slightly, in the case of the donors, upon the H3 transmission experiment (Fig 3A). Although the H1-imm recipients were protected against H3 transmission they had no detectable cross-reacting neutralizing antibodies against H3 (Fig 3A, green line). At day 97 post-infection, only the experimentally infected donors (H1-H3 INF) had neutralizing antibody titers against H3 (Fig 3A, green line).

Infection elicits a polyclonal response and antibodies targeting the immunosubdominant surface glycoprotein, neuraminidase (NA) are made in addition to those directed against HA. NA specific antibodies are able to disrupt NA activity and can be broadly cross-reactive within influenza virus subtypes [24]. To determine whether infection of H1N1pmd09 produced cross-reactive antibodies to the N2 protein, ELISA was used to test binding of primary H1N1pdm09 or H3N2-infected ferret serum against recombinant N1 or N2 proteins. All ferrets infected with H1N1pdm09 produce antibodies that recognized N1 at day 16 post-infection but no cross-reactivity to N2 above background levels was observed (S3 Fig). Similarly, H3N2-infected ferrets generated antibodies specific to the N2 protein but did not produce any antibodies that recognized N1 (S3 Fig). Taken together, these data indicate that the block in

H3N2 airborne transmission by H1N1pdm09 immunity is independent of either cross-reactive neutralizing antibodies and NA specific antibodies.

### Pre-existing immunity against influenza B virus does not block transmission of H3N2

Influenza A and B viruses are continuously co-circulating among the human population with no cross-protection between the two virus types being observed [25]. To interrogate whether innate immunity plays a role in blocking H3N2 transmission, we first infected 4 ferrets with influenza B virus (IBV) (B/Brisbane/60/2008). All animals shed virus in their nasal secretions (S4A Fig) and seroconverted by day 14 post-infection (S4B Fig). Sixty-seven days later, these ferrets were used as recipients (henceforth referred to as 'IBV-imm recipient') in an H3N2 transmission experiment. In Fig 4, H3 INF donors transmitted the virus to 3 out of 4 IBV-imm recipients (75%) (Fig 4B, orange bars). Efficient transmission of H3N2 in the presence of recipient IBV pre-existing immunity suggests that adaptive immune responses specific to IAV, rather than innate responses, are responsible for the block to H3N2 transmission in animals with H1N1pdm09 pre-existing immunity.

## Discussion

Airborne transmission is critical for emergence of pandemic viruses. We demonstrate that epidemiologically successful human seasonal and pandemic influenza viruses transmit to naïve recipients efficiently even at shortened or intermittent exposure times. In humans, primary

**A**

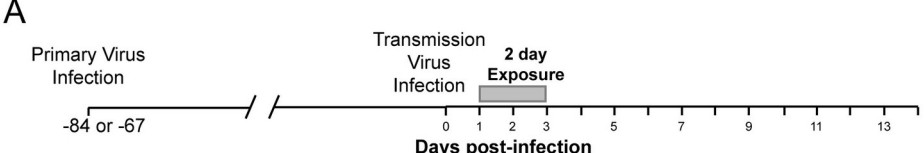

**B**

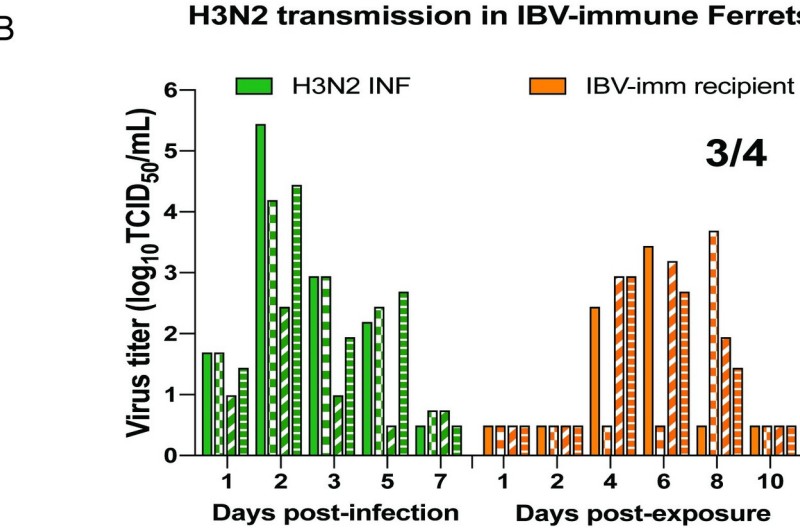

**Fig 4. Innate immunity does not block H3N2 transmission.** (A) Schematic of experimental procedure, gray bars depict 2-day exposure time. (B) Recipient animals were infected with B/Brisbane/60/2008 (IBV) virus 67 days prior to H3N2 (A/Perth/16/2009) transmission. Donor animals are denoted as H3 INF, and recipient animals are 'IBV-imm'. Each bar represents an individual animal. Limit of detection is denoted by a dashed line.

influenza infections typically occur by about 5 years of age, which initiates innate and adaptive immune responses resulting in immunological memory. Thus, the spread of pandemic and seasonal IAV occurs in a population with significant pre-existing immunity [23]. We demonstrate that pre-existing immunity can influence airborne transmissibility of IAV. Specifically, transmission of a seasonal H3N2 virus is abrogated in ferrets with pre-existing H1N1pdm09 immunity that was non-neutralizing. Conversely, H3N2 immunity did not significantly impair H1N1pdm09 transmission. Importantly, recipient immunity is sufficient to completely block H3N2 transmission through an adaptive immune mechanism that is independent of cross-neutralizing antibodies.

Transmissibility of influenza viruses relies on viral replication in the upper respiratory tract, release of virus into the air, survival of the virus within the environment, and successful replication in a susceptible recipient. In the ferret animal model, transmission studies typically house a naïve recipient animal adjacent to an infected donor for 14 consecutive days [12–15] and very few studies have altered this experimental animal model. In this study, we examined the transmissibility of seasonal H3N2 and H1N1pdm09 viruses at shortened exposures times and found efficient transmission after a 48 hour continuous exposure and intermittent exposure of 8 hours a day for 5 days, mimicking a work week. Our data with H1N1pdm09 are consistent with previously published studies demonstrating efficient transmission of H1N1pdm09 to naïve ferrets after a 30 hour exposure to as short as a 3 hour exposure [23,26,27]. Interestingly, transmissibility of H1N1pdm09 within a 30 hour exposure window was efficient if the naïve recipients were exposed to infected ferrets early (1–2 days) as opposed to late (5–6 days) post-infection [28]. Further shortening the exposure period to 8 hours and placing the naïve recipient in the adjacent cage on days 1, 3 or 5 post-infection also demonstrated that airborne transmission is more efficient before the onset of symptoms [27]. All of these studies have distinct transmission setups, with different airflow rates and cage sizes, which may impact the transmission efficiency over these short time scales. However, we can conclude that transmission efficiency is linked to exposure time.

Shortening the exposure time likely reduced the number of virus-laden aerosols inhaled by a recipient. However, the ferret infectious dose 50 ($FID_{50}$) of H1N1pdm09 is reported to be <100 plaque forming units (pfu) and the $FID_{50}$ of an H3N2 virus from 1999 (A/Panama/2007/ 1999) is 1–2 pfu [29–31]. The low $FID_{50}$ values for human H1N1 and H3N2 viruses in ferrets suggest that naïve animals are highly susceptible to these viruses. Interestingly, human dose-finding studies performed within the last decade have concluded that ~$10^7$ $TCID_{50}$ H1N1pdm09 or H3N2 viruses are needed to produce an infection in humans that results in mild to moderate symptoms [32,33]. Thus, the decreased transmission efficiency we observed could be accounted for by an increase in the $FID_{50}$ in the presence of pre-existing immunity.

In this study, we also demonstrate that heterosubtypic immunity can provide protection from transmission of H3N2 virus. A broadly protective immune response can be generated between HA subtypes [4,5,34–39], but the lack of detectable H3N2 neutralizing antibodies in the serum of donors and recipients suggests that these broadly cross-reactive neutralizing antibodies are not the source of protection. Although the primary correlate of protection against influenza virus infections has been mapped to antibodies against the hemagglutinin (HA) head domain [40], antibodies against the HA stalk domain can also be protective. H1 stalk antibodies elicited by vaccination do not neutralize H1N1pdm09 viruses but do partially protect ferrets from H1N1pdm09 infection [41]. Sub-neutralizing concentrations of antibodies that recognize the HA stalk have been shown to limit influenza disease through antibody-dependent cell mediated cytotoxicity (ADCC) [41–44]. We recently showed that HA stalk antibody titers are boosted after secondary heterosubtypic infections of ferrets [45] and it is possible that HA stalk antibodies recalled from primary infections provide some level of heterosubtypic immunity in our ferret model.

The viral NA has a critical role in virus replication and is a primary target for many licensed influenza antivirals. In guinea pigs, intranasal vaccination against IBV NA produced a robust anti-NA immunity at the mucosal surface, which caused a slight reduction in shedding by the immunized donors but more importantly blocked homologous transmission and reduced heterologous IBV transmission to naïve recipients [46]. This result is similar to data shown in Fig 2D, in which donor immunity contributes to a reduction in heterologous transmission. However, analysis of NA binding antibodies revealed that ferrets infected with H1N1pdm09 produced robust anti-N1 antibodies 16 days post-infection, but no cross-reactive antibodies targeting N2 (S3 Fig). These data suggest that NA antibodies are not the correlate of protection in our setup. Studies have shown that mucosal immunity against NA is more important than serum antibody levels at reducing transmission [46–48], thus further examination of mucosal NA antibodies may be warranted.

It is also possible that non-neutralizing antibodies against the influenza virus conserved antigens, NP, M1 and M2, which may exert a protective function through ADCC, antibody-dependent phagocytosis or antibody-mediated complement-dependent cytotoxicity [49]. Additionally, T-cell mediated immune responses targeting the highly conserved NP have been associated with cross-reactive cytotoxic T lymphocytes [50]. Future studies exploring the immunogenicity of H1N1pdm09 NP compared to the H3N2 NP may provide insight into whether the protection provided by H1N1pdm09 primary infection is driven by T-cell immunity. Examination of cross-reactive adaptive immune responses between H1N1pdm09 and H3N2 will further our understanding of influenza virus correlates of protection.

The immune response generated by influenza virus infection is a much broader and longer lasting one than that elicited by vaccination [24,39,51,52]. Yet, live vaccines are capable of inducing higher levels of mucosal immunoglobulin A (IgA) antibodies, while inactivated vaccines tend to induce a serum anti-HA antibody response [53]. It is likely that the route of immunization is important for a vaccine's ability to generate a cross-reactive immune response and thereby limit transmission. This will be an important avenue of future research.

In the last 11 years, two pandemic viruses have emerged in the human population; the 2009 pandemic H1N1 virus and SARS-CoV-2. Both viruses are efficiently transmitted from person-to-person within expelled aerosols. During the 2009 H1N1 influenza pandemic over 12,000 individuals died of the infection in the United States, with 77% of those deaths occurring in patients 18–64 year-old [3]. Recent analyses correlating birth year to imprinted virus would suggest that a large proportion of those individuals would have been imprinted with a seasonal H3N2 virus [4] and our data demonstrate that pre-existing immunity against H3N2 virus was not a significant barrier to natural infection of H1N1pdm09 virus. In contrast, in 2017–2018 a drifted H3N2 predominated for which the vaccine was not efficacious, but individuals aged 5–24 had the lowest number of infections in that year [2] and would include a large number of individuals imprinted with H1N1pdm09 virus. Previous studies in ferrets have also demonstrated that pre-existing H1N1pdm09 immunity 31 days prior allowed for rapid clearance of an antigenically distinct swine influenza virus and demonstrated reduced disease severity compared to naïve ferrets [54]. Although the immunological mechanism underlying this phenotype will require further studies, translation of these results to the current COVID-19 pandemic may be important to understand age-based distributions of SARS-CoV-2 disease severity and susceptibility.

## Methods

### Ethics statement

Ferret transmission experiments were conducted at the University of Pittsburgh in compliance with the guidelines of the Institutional Animal Care and Use Committee (approved protocol

#16077170 and #19075697). Animals were sedated with approved methods for all procedures. Isoflurane was used for all nasal wash and survival blood draw, ketamine and xylazine were used for sedation for all terminal procedures followed by cardiac administration of euthanasia solution. Approved University of Pittsburgh Division of Laboratory Animal Resources (DLAR) staff administered euthanasia at time of sacrifice.

## Cells and viruses

MDCK (Madin Darby canine kidney, obtained from ATCC) and MDCK SIAT cells (kind gift from Dr. Stacy Schultz-Cherry at St. Jude) were grown at 37˚C in 5% $CO_2$ in MEM medium (Sigma) containing 5% Fetal Bovine Serum (FBS, HyClone), penicillin/streptomycin and L-glutamine. Reverse genetics plasmids of A/Perth/16/2009 and A/California/07/2009 were a gift from Dr. Jesse Bloom (Fred Hutch Cancer Research Center, Seattle) and B/Brisbane/60/2008 was a provided by Dr. Andrew Mehle (University of Wisconsin-Madison). All viruses were rescued as previously described in [30]. The viral titers were determined by tissue culture infectious dose 50 ($TCID_{50}$) using the endpoint titration method on MDCK cells for H1N1pdm09 and IBV and MDCK SIAT cells for H3N2 [55].

## General animal details

Five to six month old male ferrets were purchased from Triple F Farms (Sayre, PA, USA). All ferrets were screened for antibodies against circulating influenza A and B viruses by hemagglutinin inhibition assay, as described in [30], using the following antigens obtained through the International Reagent Resource, Influenza Division, WHO Collaborating Center for Surveillance, Epidemiology and Control of Influenza, Centers for Disease Control and Prevention, Atlanta, GA, USA: 2018–2019 WHO Antigen, Influenza A(H3) Control Antigen (A/Singapore/INFIMH-16- 0019/2016), BPL-Inactivated, FR-1606; 2014–2015 WHO Antigen, Influenza A(H1N1)pdm09 Control Antigen (A/California/07/2009 NYMC X-179A), BPL-Inactivated, FR-1184; 2018–2019 WHO Antigen, Influenza B Control Antigen, Victoria Lineage (B/Colorado/06/2017), BPL-Inactivated, FR-1607; 2015–2016 WHO Antigen, Influenza B Control Antigen, Yamagata Lineage (B/Phuket/3073/2013), BPL-Inactivated, FR-1403.

## Transmission studies

Our transmission caging setup is a modified Allentown ferret and rabbit bioisolator cage similar to those used in [17,30]. Additional details on the caging setup can be found in S5 Fig. For each study, three to four ferrets were anesthetized by isoflurane and inoculated intranasally with $10^6$ $TCID_{50}$/500uL of A/Perth/16/2009, A/California/07/2009 or B/Brisbane/60/2008 to function as the donor (INF) animals or to generate pre-existing immunity in recipient animals. Twenty-four hours later, a recipient ferret was placed into the cage but separated from the donor animal by two staggered perforated metal plates welded together one inch apart. The recipient was removed from the cage after 7 or 2 days and then individually housed. During the intermittent exposure, recipients were removed from the cage every 8 hours and individually housed in a new cage every day for 5 consecutive days. Nasal washes were collected from each donor and recipient every other day for 14 days. To prevent accidental contact or fomite transmission by investigators, the recipient ferret was handled first and extensive cleaning of all chambers, biosafety cabinet, and temperature monitoring wands was performed between each recipient and donor animal and between each pair of animals which also included glove and anesthesia chamber changes. Sera was collected from donor and recipient ferrets upon completion of experiment to confirm seroconversion. Environmental conditions were monitored daily and ranged between 20–22˚C with 44–50% relative humidity. To ensure no

accidental exposure during husbandry procedures, recipient animal sections of the cage were cleaned first then then infected side, three people participated in each husbandry event to ensure that a clean pair of hands handled bedding and food changes. One cage was done at a time and a 10 minute wait time to remove contaminated air was observed before moving to the next cage. New scrappers, gloves, and sleeve covers were used on subsequent cage cleaning.

Clinical symptoms such as weight loss and temperature were recorded during each nasal wash procedure and other symptoms such as sneezing, coughing, lethargy or nasal discharge were noted during any handling events. Animals were given A/D diet twice a day to entice eating once they reached 10% weight loss. A summary of clinical symptoms for each study are provided in S1 Table.

## RNA harvest and quantification of viral load

Viral RNA was extracted from ferret nasal washes by processing 200 μl through the Purelink Pro 96 Viral RNA/DNA Purification Kit (Thermofisher 12280-096A). The viral load in each sample was measured by RT-qPCR using primers/probe specific for the open reading frame of segment 7 (M1/M2) of influenza A virus: forward primer 5'- GACCRATCCTGTCACCTCT GAC-3', reverse primer 5'- AGGGCATTYTGGACAAAKCGTCTA-3', and probe 5'-(FAM)-TGCAGTCCTCGCTCACTGGGCACG-(BHQ1)-3'. Each reaction contained 5.4 μL of nuclease-free water, 0.5 μL of each primer at 40 μM, 0.1 μL of ROX dye, 0.5 μL SuperScript III RT/Platinum Taq enzyme mix, 0.5 μL of 10 μM probe, 12.5 μL of 2x PCR buffer master mix, and 5 μL of extracted viral RNA. The PCR master mix was thawed and stored at 4˚C, 24 hours before reaction set-up. The RT-qPCR was performed on a 7500 Fast real-time PCR system (Applied Biosystems) with a machine protocol of 50˚C- 30min, 95˚C-2min followed by 45 cycles of 95˚C-15sec, 55˚C-30sec. To relate genome copy number to Ct value, we used a standard curve based on serial dilutions of a plasmid control, run in triplicate on the same plate. H3N2 samples were compared to a plasmid containing the M segment of A/Perth/16/2009. H1N1 samples were compared to a plasmid containing the M segment of A/California/07/2009.

## Serology assay

Analysis of neutralizing antibodies from ferret sera was performed as previously described [30]. Briefly, the microneutralization assay was performed using $10^{3.3}$ $TCID_{50}$ of either H3N2 or H1N1pdm09 virus incubated with 2-fold serial dilutions of heat-inactivated ferret sera. The neutralizing titer was defined as the reciprocal of the highest dilution of serum required to completely neutralize infectivity of $10^{3.3}$ $TCID_{50}$ of virus on MDCK cells. The concentration of antibody required to neutralize 100 $TCID_{50}$ of virus was calculated based on the neutralizing titer dilution divided by the initial dilution factor, multiplied by the antibody concentration.

## ELISA

96-well ELISA plates (Immulon) were coated with 50uL of 1 μg/ml recombinant N1 or N2 protein (BEI Resources NR-19234 and NR-43784) and incubated overnight at 4˚C. Plates were blocked for 1 h at room temperature with 150 μL PBS with 0.01% Tween-20, 3% normal goat serum and 3% milk powder. Ferret sera was heat inactivated at 56˚C for 30 minutes. Two-fold serial dilutions were performed in a blocking buffer and incubated on the ELISA plate for 2 h at room temperature. After washing with PBS-0.01% Tween-20, plates were incubated with peroxidase-conjugated goat anti-ferret IgG (Abcam ab112770). SureBlue TMB peroxidase substrate (KPL) was added to each well and the reaction stopped with 250 mM HCl. Absorbance was read at 450 nm.

## Supporting information

**S1 Fig. Transmission of H3N2 to naïve ferrets for a short exposure period.** Three ferrets were infected with A/Perth/16/2009 (H3N2) and nasal washes were collected from each ferret on the indicated days post-infection. A naïve ferret was placed in the adjacent cage at 24 hour post-infection for 2 days and nasal washes were collected from each recipient ferret on the indicated days post-exposure. Bars indicate individual ferrets. All ferrets were serologically negative for circulating influenza viruses at the beginning of the study. The limit of detection was $10^{0.5}$ TCID$_{50}$/mL. TCID$_{50}$, 50% tissue culture infectious dose. The viral titer data for 3 donor animals was previously published in (45).
(TIF)

**S2 Fig. Quantification of H1N1 and H3N2 viral RNA in donor animals.** RNA was isolated from nasal wash samples at each of the indicated days post-infection from ferrets infected with H1N1pdm09 (red line) and H3N2 (green line). Data are shown as mean +/- SEM for 3–4 ferrets per condition.
(TIF)

**S3 Fig. Primary influenza virus infection does not produce cross-reactive NA antibodies.** Ferrets were infected with either H1N1pdm09 (red) or H3N2 (green) and NA antibody levels were determined by ELISA using (A) recombinant A/California/07/2009 (H1N1)pdm09 N1 or (B) recombinant A/Brisbane/10/2007 N2 proteins. OD values for day 0 and day 14 or day 16 serum are displayed and each line indicates an individual ferret. Day 0 serum is presented in a lighter shade and with open circles, while day 14 or day 16 serum is a darker shade and solid circles.
(TIF)

**S4 Fig. Replication of IBV in ferrets.** (A) Four ferrets were infected with B/Brisbane/60/2008 and nasal washes were collected from each ferret on the indicated days post-infection. Bars indicate individual ferrets. The limit of detection is represented by the dashed line. (B) All ferrets were confirmed to be serologically negative for circulating influenza A and B viruses at the beginning of the study. The presence of influenza B antibodies on day 14 post-infection was detected in all infected animals by HAI, but neutralization titers were only observed in 2/4 infected ferrets. TCID$_{50}$, 50% tissue culture infectious dose. HAI, hemagglutination inhibition. MN, microneutralization.
(TIF)

**S5 Fig. Transmission cage setup.** A. Diagram of ferret transmission unit. B. Schematic of ferret experimental setup with constant air flow passing from infected (INF) donor to recipient ferret. The rack has a flow rate of 40 cubic feet per minute (CFM), for 35 air changes per hour within the total rack. INF donor and recipient are separated by a stainless steel divider, which is made up of two perforated plates with 5mm diameter holes, the plates are welded together 2 cm apart such that the holes are staggered.
(TIF)

**S1 Table. Clinical signs and symptoms.**
(DOC)

## Acknowledgments

The following reagents were obtained through BEI Resources, NIAID, NIH: H1 Hemagglutinin (HA) protein with C-terminal histidine tag from influenza virus, A/California/07/2009

(H1N1)pdm09, recombinant from baculovirus, NR-42635 and N2 neuraminidase (NA) protein with N-terminal histidine tag from influenza virus, A/Brisbane/10/2007 (H3N2), recombinant from baculovirus, NR-43784. We would like to thank the University of Pittsburgh DLAR technicians and veterinarians for their hard work and dedication as well as the members of Lakdawala lab for constructive discussions.

## Author Contributions

**Conceptualization:** Valerie Le Sage, Scott E. Hensley, Seema S. Lakdawala.

**Data curation:** Elizabeth E. McGrady.

**Formal analysis:** Valerie Le Sage, Jennifer E. Jones, Adam S. Lauring.

**Funding acquisition:** Scott E. Hensley, Seema S. Lakdawala.

**Investigation:** Valerie Le Sage, Jennifer E. Jones, Karen A. Kormuth, Eric Nturibi, Gabriella H. Padovani, Andrea J. French, Annika J. Avery, Richard Manivanh, Amar R. Bhagwat.

**Methodology:** Valerie Le Sage, Jennifer E. Jones, William J. Fitzsimmons, Claudia P. Arevalo, Seema S. Lakdawala.

**Project administration:** Valerie Le Sage.

**Supervision:** Seema S. Lakdawala.

**Writing – original draft:** Valerie Le Sage, Seema S. Lakdawala.

**Writing – review & editing:** Valerie Le Sage, Seema S. Lakdawala.

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
