## [Decision Letter · Decision Letter 0]

14 Sep 2020

Dear Dr. Lakdawala,

Thank you very much for submitting your manuscript "Pre-existing immunity provides a barrier to airborne transmission of influenza viruses" for consideration at PLOS Pathogens. As with all papers reviewed by the journal, your manuscript was reviewed by members of the editorial board and by several independent reviewers. In light of the reviews (below this email), we would like to invite the resubmission of a significantly-revised version that takes into account the reviewers' comments.

The reviewers all expressed enthusiasm for the study but there were questions raised about the apparent selectivity of heterosubtypic protection for the H3 viruses. While in-depth mechanistic studies in the ferret model are obviously not possible, it would be helpful to add substantial discussion on this point and/or any in vitro data you have that might address why these viruses would be a better target of these heterosubtypic immune responses. Is it something about the target cells for initial infection or replication (in mice, H3s tend to target the URT more)--is it replication rates or antigen presentation? Again, a definitive answer might not be possible but any suggestive data or discussion would be very welcome. 

We cannot make any decision about publication until we have seen the revised manuscript and your response to the reviewers' comments. Your revised manuscript is also likely to be sent to reviewers for further evaluation.

Sincerely,

Paul G. Thomas

Associate Editor

PLOS Pathogens

Ana Fernandez-Sesma

Section Editor

PLOS Pathogens

Kasturi Haldar

Editor-in-Chief

PLOS Pathogens

orcid.org/0000-0001-5065-158X

Michael Malim

Editor-in-Chief

PLOS Pathogens

orcid.org/0000-0002-7699-2064

Reviewer's Responses to Questions

**Part I - Summary**

Reviewer #1: In this article, Le Sage and colleagues evaluate seasonal influenza virus transmissibility in the ferret model. In efforts to more closely recapitulate ‘real-world’ infections, authors modulate both the prior immune history of the ferrets (by employing ferrets with pre-existing immunity to either H3N2 or H1N1 viruses) and the duration of exposure between donor and contact animals (employing 7-day, 2-day, and multi-day interrupted exposure events). Collectively, these studies support that prior infection with influenza viruses can reduce the susceptibility of contact animals to heterosubtypic influenza virus infection, independent of the presence of neutralizing antibodies in contact animals. Recreating the complex immunological history of humans in mammalian models can be a challenge, and there is a need for additional study of how preexisting immunity (developed following prior natural infection and/or vaccination) modulates virus transmissibility in mammals. While the results are not especially surprising, the underlying mechanism(s) responsible for subtype-specific differences are not determined here, and there remains a need for these types of studies to be conducted in ferrets that have experienced multiple infection/vaccination events to more closely mimic humans, the findings are still of interest to the field and represent a timely addition to the literature on this topic. Experiments are rationally designed and clearly presented, and the writing is clear and well-referenced. However, there are areas in the manuscript that would benefit from additional clarity and contextualization.

Reviewer #2: The manuscript by Le Sage et al uses the ferret model to understand the effect of pre-existing immunity on influenza A airborne transmission. The authors firstly demonstrate that airborne transmission of a seasonal H3N2 and the 2009 pandemic H1N1 virus to naïve recipients can occur efficiently even at short or periodic exposure windows. The authors then demonstrate that heterosubtypic (H1/H3) immunity can limit the transmission of H3, but less so for H1 viruses. The authors demonstrate that this is independent of neutralizing antibodies, as well as innate immunity by using a primary IBV infection. Overall, the study provides evidence for the role of pre-existing heterosubtypic immunity in limiting airborne transmission of influenza A viruses that I believe would be of interest to the readership of PLOS Pathogens and the field of influenza virology and immunology.

Reviewer #3: The authors report on the impact of previous influenza immunity to provide protection against infection by airborne transmitted influenza virus. For these studies, ferrets were infected with specific influenza A or B viruses prior to heterosubtypic infection by airborne transmitted virus. The current model is based on high inoculum infection of the donor ferrets to initiate the transmission study. The high inoculum likely overwhelmed the potency of the protective immunity established by prior infection. The novelty of the studies is weakened by previous publications from additional animal models of influenza, and the minimal pre-existing immunity induced by a single exposure prior to infection, which does not reflex the complex exposure history of most individuals to influenza infection or vaccination.

**Part II – Major Issues: Key Experiments Required for Acceptance**

Reviewer #1: no major issues identified.

Reviewer #2: My major concern is with regards to the lack of a potential mechanism underlying the limited transmission in immune animals, and why that differs between H1 and H3 infection/transmission events. The lack of neutralising antibodies between H1 and H3 viruses is not particularly surprising, neither is the lack of any innate effects, given that long period between primary and secondary infections. Any additional data (eg HA stalk titers, NA antibodies) that may be available could provide additional insights into this. At the very least, I think the discussion could benefit from some additional points on what maybe limiting transmission. It is pertinent to also mention prior evidence for:

- HA stalk antibodies can reduce susceptibility to transmission (eg ref 36 of manuscript)

- NA antibodies can limit transmission of IBV in a guinea pig model (MacMahon et al 2019, 10.1128/mBio.00560-19)

- CD8+ T cells are known to confer broad cross-protection between heterosubtypic infections and may explain some of the observations of reduced disease severity as mentioned in discussion lines 278-280.

Reviewer #3: The current studies are technically sound; however, the high infection dose likely overwhelmed the potency of pre-existing immunity. The authors should discuss whether prior reports provide evidence of cross-protective cellular immunity. Pre-existing immunity induced by IBV infection was a good control experiment, but it does not provide suggestive data as to why H3N2 prior immunity failed to protect against infection by airborne transmitted H1N1 influenza virus. It is also unclear how efficiently IBV replicated in ferrets, and thus established pre-existing immunity.

**Part III – Minor Issues: Editorial and Data Presentation Modifications**

Reviewer #1: -line 78, would this repeated exposure not be inclusive of both infection and vaccination? Overall throughout the study, it would be of benefit for the authors to clarify and expand upon if they believe vaccination would elicit comparable results, or if their findings are specific to immunity acquired via live virus infection.

-line 110, “a representative seasonal H3N2 influenza virus, A/Perth/16/2009” – how antigenically related is this strain to the currently circulating virus clades?

-Figure 1A/S3, it’s currently unclear if the contact ferrets that had intermittent exposure to donor ferrets (8hr/day for 5 days) were removed from their cage outright outside of these exposure windows to a clean cage that was free of virus, or if a solid barrier was placed at the perforated stainless steel interface separating inoculated and donor ferrets to abrogate exposure to new aerosols shed from donor ferrets (while maintaining exposure to any virus still present in the cage), please specify.

-Lines 175-78/296-8: Do the authors know the relative 50% ferret infectious dose for the H1N1 and H3N2 viruses employed here? Is it possible that higher levels of exhaled infectious virus from the H1N1 donor ferrets and/or increased transmissibility at lower exposure levels is contributing to the subtype-specific profiles of transmission presented in experiments from Figure 2 (i.e. that the H1N1 but not H3N2 virus is overcoming preexisting heterosubtypic immunity in part because of higher viral exposures in the contact animals)?

-Lines 190-92: Nasal washes capture relative levels of virus replication in the ferret respiratory tract but are not necessarily a strong surrogate for virus shed into the environment from inoculated animals; consider changing “…not due to significantly decreased viral shedding” to “…viral replication” (or otherwise clarify the statement to account for this).

Reviewer #2: - The manuscript might benefit from adding ‘heterosubtypic’ to the title. eg “Pre-existing heterosubtypic immunity provides a barrier….”

- The abstract states (lines 37-38) “viruses are highly transmissible to naïve animals within 48 hours”. This may be misinterpreted to mean that the viruses were sown to be transmitted within the first 48 hours of infection, while the 48 hour window was from 24-72 hours according to fig 1A. Can this statement be rephrased to something like “a 48 hour exposure is sufficient for efficient transmission”

- Sup table 1 states that animals in fig 2B were exposed for 7 days, while the schematic in fig 2A shows a 2 day exposure. Can the authors please clarify and amend accordingly.

Reviewer #3: How pre-existing immunity against infection by transmitted influenza virus is a potential correlate of protection is unclear and should be explained with more detail. The authors indicated prior studies that examined this topic, but it is unclear how the current experimental design improves upon these previous studies. The authors indicated differences between air flow rates, but it is unclear if liters per minute are similar to or substantially divergent from cubic feet per minute.

PLOS authors have the option to publish the peer review history of their article (what does this mean?). If published, this will include your full peer review and any attached files.

Reviewer #1: No

Reviewer #2: No

Reviewer #3: No
---

## [Decision Letter · Decision Letter 1]

4 Jan 2021

Dear Dr. Lakdawala,

We are pleased to inform you that your manuscript 'Pre-existing heterosubtypic immunity provides a barrier to airborne transmission of influenza viruses' has been provisionally accepted for publication in PLOS Pathogens.

Before your manuscript can be formally accepted you will need to complete some formatting changes, which you will receive in a follow up email. A member of our team will be in touch with a set of requests. One reviewer suggested another thorough read for editorial proofreading would be helpful as well.

Best regards,

Paul G. Thomas

Associate Editor

PLOS Pathogens

Ana Fernandez-Sesma

Section Editor

PLOS Pathogens

Kasturi Haldar

Editor-in-Chief

PLOS Pathogens

orcid.org/0000-0001-5065-158X

Michael Malim

Editor-in-Chief

PLOS Pathogens

orcid.org/0000-0002-7699-2064

Reviewer Comments (if any, and for reference):

Reviewer's Responses to Questions

**Part I - Summary**

Reviewer #2: The authors have addressed the concerns raised by the reviewers

Reviewer #3: The manuscript submission by Lakdawala and colleagues reports on observations from the ferret model of influenza that pre-existing heterosubtypic immunity may govern airborne transmission of influenza A viruses. Specifically, the results from the ferret model indicates that pre-existing H1N1 influenza virus immunity reduces airborne transmission of H3N2 influenza virus, but the converse does not apply. The authors speculate on the immunological mechanisms, but necessary experiments are beyond the scope of the current study. There is potential clinical relevance of the current findings such that pre-existing influenza immunity may reduce influenza virus transmission dynamics among individuals with pre-existing immunity. This observation may help explain the variation in the influenza subtypes that predominate during annual epidemics of influenza. The revised manuscript addressed the previous concerns of this reviewer.

**Part II – Major Issues: Key Experiments Required for Acceptance**

Reviewer #2: (No Response)

Reviewer #3: No major issues requiring additional experiments.

**Part III – Minor Issues: Editorial and Data Presentation Modifications**

Reviewer #2: (No Response)

Reviewer #3: The manuscript requires minor proofreading.

PLOS authors have the option to publish the peer review history of their article (what does this mean?). If published, this will include your full peer review and any attached files.

Reviewer #2: No

Reviewer #3: No

---

## [Editor Report · Acceptance letter]

25 Jan 2021

Dear Dr. Lakdawala,

We are delighted to inform you that your manuscript, "Pre-existing heterosubtypic immunity provides a barrier to airborne transmission of influenza viruses," has been formally accepted for publication in PLOS Pathogens.

Best regards,

Kasturi Haldar

Editor-in-Chief

PLOS Pathogens

orcid.org/0000-0001-5065-158X

Michael Malim

Editor-in-Chief

PLOS Pathogens

orcid.org/0000-0002-7699-2064